# Effectiveness of Multimedia-Based Learning on the Improvement of Knowledge, Attitude, and Behavioral Intention toward COVID-19 Prevention among Nurse Aides in Taiwan: A Parallel-Interventional Study

**DOI:** 10.3390/healthcare10071206

**Published:** 2022-06-28

**Authors:** Yi-Min Hsu, Ting-Shan Chang, Chien-Lun Chu, Shu-Wen Hung, Chih-Jung Wu, Tzu-Pei Yeh, Jiun-Yi Wang

**Affiliations:** 1Department of Nursing, China Medical University Hospital, No. 2, Yude Rd., North District, Taichung 404332, Taiwan; n4006@mail.cmuh.org.tw (Y.-M.H.); n32638@mail.cmuh.org.tw (S.-W.H.); 2School of Nursing, China Medical University, No. 100, Sec. 1, Jingmao Rd., Beitun District, Taichung 406040, Taiwan; tzupeiyeh@mail.cmu.edu.tw; 3School of Nursing, Fooyin University, 151 Jinxue Rd., Daliao District, Kaohsiung 83102, Taiwan; u104077804@cmu.edu.tw; 4Cancer Registry and Screening, Cancer Center, China Medical University Hospital, No. 2, Yude Rd., North District, Taichung 404332, Taiwan; chienlun0712@gmail.com; 5Department of Hematology and Oncology, China Medical University Hospital, No. 2, Yude Rd., North District, Taichung 404332, Taiwan; elvaamy@gmail.com; 6Department of Healthcare Administration, Asia University, 500, Lioufeng Rd., Wufeng, Taichung 41354, Taiwan

**Keywords:** COVID-19, multimedia-based learning, nurse aides

## Abstract

Because nurse aides are one of the first-line care providers in hospitals, they should possess better knowledge, attitude, and behavioral intention toward COVID-19 during the pandemic. This study aimed to compare the improvements of COVID-19-related education on learning outcomes between multimedia-based and traditional face-to-face learning models for nurse aides. The parallel-group randomized controlled trial recruited 74 participants in both the experimental and control groups. Two 90 min interventions with the same contents, but in different ways, were delivered. A structured questionnaire was used to collect data of demographic information, knowledge, attitude, and behavioral intention toward COVID-19 before and after the interventions. Results from generalized estimation equations analysis indicated that the nurse aides in the multimedia-based learning group had greater improvement in the scores of knowledge (difference in change: 3.2, standard error: 0.97, *p* < 0.001), attitude (difference in change: 10.2, standard error: 2.97, *p* < 0.001), and behavioral intention (difference in change: 0.5, standard error: 0.04, *p* < 0.001) than those in the face-to-face learning group. During the outbreak of COVID-19, multimedia-based learning as an effective learning method could improve the learning outcomes related to COVID-19 and achieve learning goals without close contact.

## 1. Introduction

In the end of 2019, Wuhan City in the Hubei province of China witnessed an upward spiral in the number of pneumonia cases with a new strain of coronavirus [1], namely COVID-19. The rapid spread of infections from the new strain has engendered a global pandemic. Regardless of the fact that SARS-CoV caused severe acute respiratory syndrome from 2002 to 2003 and MERS-CoV triggered serious respiratory diseases in 2012 [2,3], medical communities in many countries still lack the experience to deal with the COVID-19 outbreak. Moreover, currently there is absence of any effective medication for this virus, and vaccines have not yet been comprehensively used worldwide. If governments and health authorities fail to adopt necessary infection control approach in a timely manner, it is very likely to see a rapid incidence of community infection, which can lead to exacerbated medical costs and fatalities [4,5].

The presence of COVID-19 cases with no symptoms or mild symptoms makes infection control more difficult. Without confirmation whether a patient is infected or not, healthcare providers are prone to contract the virus and can even be infected [6]. Lack of accurate knowledge of the infectious disease is common to lead fear, misunderstanding, and distort views in the general public, causing avoidance and discrimination toward infected people or healthcare providers [7]. Thus, providing relative education to healthcare providers of COVID-19 infection prevention and methods to abate the transmission rate becomes particularly critical.

Aging population has been increasing in Taiwan. Nurse aides have become the primary workforce of caregiving in hospitals, home care services, and long-term care institutes in Taiwan. For long-term care cases, they have the highest risk of having critical symptoms and fatality among the population who have COVID-19 [8]. During the COVID-19 outbreak, it is unavoidable that nurse aides may have to care infected patients, and it is imperative to strengthen their knowledge and attitude toward COVID-19. However, even if there are urgent needs for nurse aides to improve their knowledge and skills of infection control, it could be challenging to follow learning schedules during the pandemic, and this may impact their ability to receive on-the-job education and training. Factors related to nursing staffs’ willingness to attend continuous in-service education include the shortage of workforce, prolonged working time, and work-life imbalances. Traditional lecture methods of in-service education could be difficult for nurses mainly due to working time and lecture schedule conflicts, as well as travel. Through offering digital learning, the difficult issues described above could be addressed [9].

On-line education is not a novel concept; human society began a whole new era in learning a long time ago because of the boost of digital world [10]. Multiple media used in education could resolve past difficulties such as limitations in time and location; this method allowed more and more learners to obtain crucial knowledge without the above concerns. Theories in digital learning include behaviorism, cognitivism, and constructivism. Behaviorism helps educators to make the curriculum design structured so that target learners can handle the concepts, skills, and relative information clearly; in addition, according to these constructions, more appropriate methods will be used to evaluate learning outcomes. Regarding constructivism, it could be used to understand the learners’ experiences of constructing individual knowledge [11,12,13]. In terms of teaching strategies applied in digital learning, the strategies focus on learner-based methods; the teacher should be leaders and facilitators. Students are encouraged to become self-regulated, self-mediated, and self-aware; therefore, they may actively choose learning targets and control the progress of learning. Creativity problem solving teaching models could be used simultaneously for enhancing stimuli, knowledge internalization, and coping diversity; this circumstance may push students to develop abilities in independent judgment and logical thinking. Multimedia-based learning also emphasizes the influences of contexts, especially learning scenarios and groups; for instance, interactions between peers or students and teachers. Cooperation learning is recognized as a great learning strategy to make learning meaningful and increase the learning effects [14]. However, multimedia-based learning has its limitations, including whether sound teaching and evaluation platforms are available, the learners’ motivations and activity, and the teaching scenario settings. As long as these limitations can be avoided, the multimedia-based learning may reach maximum effectiveness.

Studies have indicated that multimedia learning has shown beneficial outcomes, such as effective delivery of information to learners, attention attraction, better understanding among learners of learning contents, and the improvement of professional awareness and skills among healthcare providers. Therefore, this method is widely used to deliver health-related education in clinical settings [15,16,17]. 

After the COVID-19 outbreak, to avoid the incidences of cluster infection, digital-based learning has become the priority learning method in schools and hospitals. However, more evidence would be required to confirm the benefit of digital-based learning under COVID-19. This study aimed to explore and compare the learning outcomes between the multimedia-based learning and traditional face-to-face learning regarding the knowledge, attitude, and behavioral intention of nurse aides. Nurse aides are assumed to achieve better understanding in regard to COVID-19-related issues through multimedia-based education, so that they can have better self-protection and management which may result in higher willingness to take care of infected cases.

## 2. Materials and Methods 

### 2.1. Study Design and Procedure

This is an interventional study with two parallel groups. Participants who worked in a medical center in central Taiwan were enrolled in March 2020. The participants were nurse aides with full training qualifications in Taiwan and with at least six months of work experience. This study adopted random grouping to allocate nurse aides from this medical center into two groups by lots. The participants in the experimental group received teaching intervention delivered by multimedia-based teaching, and the others who were in the control group received traditional face-to-face teaching. All participants were asked to complete a pre-test questionnaire at 90 min before the learning intervention and a post-test questionnaire. The two groups were not allowed to discuss or use books or websites when they were filling in the questionnaires. Everyone sat at a distance to maintain social distance. The number of participants was 74 in each group. Among the 148 participants, 21 of them (16 from the experimental group and 5 from the control group) were excluded from the study because of uncompleted responses in the questionnaires. Ultimately, the analyzed sample included 69 and 58 participants in the experimental and control groups, respectively (Figure 1).

### 2.2. Ethics Approval

This study was approved by the Institutional Review Committee of China Medical University in Taiwan (No: CMUH109-REC1-056). The purpose and procedures of this study were explained to the potential participants before enrolment. After obtaining informed consent forms, those participants were introduced and started the training modules.

### 2.3. Educational Intervention Program

Participants in the experimental group received educational intervention delivered by multimedia-based learning. The education material was mainly formed from the policies issues of the Central Epidemic Command Center (CECC) in Taiwan and reviewed by the infection control team in the hospital. After careful revision, the material was edited, designed, and subsequently recorded as a video for educational use. The education contexts included COVID-19-related knowledge and necessary skills demonstrations. The course contexts are: (1) knowledge of COVID-19; (2) hand hygiene; (3) steps to put on and take off protective equipment; and (4) cough etiquette, etc. The education materials were reviewed and assessed for practical feasibility by five specialists, including an infection control physician, a thoracic medicine physician, an infection control nurse, a head nurse of the infection ward, and a university professor. Based on the professionals’ consensus and feedback on the first version of the education material, the video was revised accordingly to ensure the appropriateness and accuracy of the contents. In the next phase, one nurse aide was invited to view the video to ensure that the multimedia-based intervention was feasible and acceptable. The COVID-19 prevention video was provided to every participant in the experimental group. As for the control group, the intervention was delivered in a traditional face-to-face approach, and an infection control nurse delivered the lecture in which the content was kept the same as the experimental group. The duration of the intervention for both groups lasted for 90 min. Therefore, the comparison of the learning outcomes from different intervention methods could be evaluated fairly.

### 2.4. Instruments

This study applied a self-developed questionnaire combined the research from Alkot (2016) and information from the researchers’ clinical experiences. The questionnaire includes demographic information, scales of knowledge, attitude, and behavioral intention toward COVID-19. Demographic information includes age, gender, educational levels, work duration as a nurse aide (years), experience in providing care for cases with infectious diseases, and experience in providing care for suspected COVID-19 cases. The knowledge scale consists of 11 items (including the incubation period, the transmission route, etc.). Each correct answer gets one point. The total scores range from 0 to 11, a higher score represents possessing better knowledge of COVID-19. This scale shows good reliability and validity; the average value of the content validity index (CVI) from six experts was 0.92, and the value of Kuder–Richardson formula 21 (KR21) was 0.523. The attitude scale comprises 15 items (including fear of becoming a source of infection or being infected, fear of going to a hospital to work, etc.). The five-point Likert scale was applied; thus, the total scores range from 15 to 75. A higher score indicates more positive attitude toward COVID-19. The scale also exhibits good reliability and validity; the average value of the CVI from six experts is 0.94, and the value of Cronbach’s alpha is 0.764. The behavioral intention scale is composed of six items (including the willingness to comply with protective measures and to avoid crowded places, etc.). An answer following the suggested guideline gets one point. The total scores range from 0 to 6; a higher score represents a more positive behavioral intention toward COVID-19. This scale has good reliability and validity as well; the average value of the CVI = 0.97, and the value of KR21 = 0.507.

Sample size was calculated by using the G*power 3.1 software [18]. When the sample size is 98–150, two-group repeated measures ANOVA (Analysis of Variance) was set with a 0.05 significance level and had 80% power to detect a difference in means across the three levels of the repeated measures factor characterized by a correlation among repeated measures of 0.5 and an effect of 0.2–0.25.

### 2.5. Statistical Analysis

After data was cleaned, statistical analyses were performed by using IBM SPSS (Statistical Product and Service Solution) version 22. For descriptive analysis, the number of participants, percentage, mean, and standard deviation were used to present the distributions of the variables. Score change in each scale was defined as the score in the post-test minus the score in the pre-test. For inferential analysis, the independent t-test, chi-square test, one-way ANOVA, and post hoc analysis were used to examine the difference of the means between the two groups; multiple linear regression was done to determine which variables in the demographic characteristics were related to the score changes in knowledge, attitude, and behavior pattern. The Pearson correlation coefficient was analyzed to measure the associations of the score changes in the knowledge, attitude, and behavior patterns in nurse aides. Finally, the generalized estimating equations (GEE) were used to evaluate the effects of the two teaching methods on the score changes. The significance level of the statistical analysis was set to 0.05.

## 3. Results

Among the 127 study participants, the majority were female (74.8%), over 50 years old (79.5%), and married (81.1%). In terms of educational levels, 51.2% of the participants held a senior high school degree, and 37% had a junior high school degree. Most of the participants had worked for three to ten years (45.6%), followed by those for more than ten years (40.2%); 62.5% of the participants had experience in providing care for patients with infectious diseases, only 0.4% of them had experience in providing care for cases with COVID-19. The results of chi-square test demonstrated that there were significant differences between the two groups in the variables of age (*p* = 0.013), years of working (*p* < 0.001), the experience of caring for patients with infectious diseases (*p* = 0.016), and the experience of caring for patients with COVID-19 (*p* = 0.042). No significant difference was found in genders, marital statuses, and educational levels (Table 1).

Based on the paired t-test, it showed that the scores in the post-test were significantly higher than those in the pre-test in both groups in all three scales (*p* < 0.001). The interactions in the generalized estimation equations (GEE) also showed significant results (*p* < 0.001), which meant the nurse aides in the multimedia-based learning group had greater improvement in knowledge, attitude, and behavioral intention than those in the face-to-face learning group (Table 2).

The correlation analysis showed significant results between either two of knowledge, attitude, and behavioral intention (*p* < 0.01), in which the correlation coefficients were 0.288, 0.534, and 0.235, respectively; the greater the score changes of knowledge, the greater change can also be found in the score of attitude, and likewise to the score of behavioral intention (Table 3).

Results of the regression analysis suggested that educational level was an associated factor, where participants holding a junior high school degree had greater score improvement than those holding junior college or university degree (t = −2.375, *p* < 0.05). Apart from that, participants with experience of caring for COVID-19 cases had greater score improvement in the knowledge dimension (t = −3.588, *p*< 0.001). In terms of the associated factors for the mean score changes of attitude, participants who had more than ten years of work experience had greater improvement than those who had been worked less than three years (t = 2.362, *p* < 0.05). Regarding the score improvements of knowledge and attitude, the traditional face-to-face learning group showed smaller score changes than the multimedia-based learning group (both *p* < 0.001). In addition, the score improvement of behavioral intention was significantly associated with the score improvement of knowledge (*p* < 0.001) (Table 4).

Score change was defined as the score in the post-test minus the score in the pre-test.

## 4. Discussion

The result of this study showed that in the multimedia-based learning group, the scores of knowledge, attitude, and behavioral intention in the nurse aides in the post-test are significantly higher than those in the pre-test. Educational levels, working durations, and willing to take care of COVID-19 patients are significant related variables to nurse aides’ knowledge, attitude, and behavior. The research findings indicated that the higher educational level group revealed the higher scores in knowledge, which is consistent with previous studies abroad [19,20]. Naser et al. (2020) conducted a survey across three Middle Eastern countries (Jordan, Saudi Arabic, and Kuwait) under the COVID-19 pandemic, to understand people’s knowledge and behavior. In 1208 Middle Eastern people, in those who had educational levels, their scores in COVID-19-related knowledge were higher [21]. Chang, Tzou, and Rehn (1999) investigated 60 Taiwanese nurses’ knowledge and behavior intention toward Acquired Immune Deficiency Syndrome (AIDS); the results indicated that nurses who had more work experience, were older in age, and had had experiences in caring for AIDS patients showed the higher scores in knowledge dimension. The research results were similar to this study. Those nurse aides who had more work experience and had actual caring experience in caring for infectious disease patients showed higher levels in related knowledge. Regarding the older ages and working durations in nurse aides, they may have more opportunities to receive educational training; therefore, in this study, nurse aides who had longer working durations obtained higher scores in knowledge. According to previous literature, knowledge may directly influence attitude; when medical and nursing staffs have richer knowledge, they also have greater confidence to beat the virus [22]. The learners’ self-motivation is an important factor in multimedia-based learning [23]. This study focused on COVID-19 prevention education in nurse aides, which may influence their own safety to work in hospitals during the pandemic. In addition, nurse aides mainly gain new knowledge of disease care via continuous in-service education from hospitals. Therefore, the motivation of the learners was quite sufficient in this study. 

Chang and Hsu (2010) conducted research on nurses by using multimedia as a learning method in Taiwan; the results showed that the scores of the electrocardiogram (ECG) knowledge of the experimental group were higher than the control group [24]. This finding suggested that digital-based learning strategies can be utilized in continued training for nursing staff to increase the interest and motivation in learning, and also in improving the learning results. Yueh et al. (2012) examined the attitudes and perspectives of graduate students toward lectures delivered through multimedia, and the result indicates that the students preferred the multimedia-assisted learning method instead of traditional lectures in classrooms. Moreover, digital-based learning is beneficial in improving students’ comprehension of course contents; thus, the learning ability is increased [25]. A study by Hemmati, Omrani, and Hemmati (2013) also showed a similar result. In their study, they delivered continuing medical education of CPR training to 80 physicians, where the training was conducted in two models—digital-based learning and traditional classroom lecture with slides. After the training sessions, tests were conducted, and the results indicated that the score of the digital-based learning was higher than the traditional slideshows lectures [26]. George et al. (2014) conducted a systematic review which included 59 articles; the study participants included 6750 students in medicine, dentistry, nursing, physical therapy, and pharmacology. The study results revealed that the students who received lectures in a digital model had significantly better outcomes in knowledge acquirement, attitude, and satisfaction than those who received lectures in classrooms [27]. Momennasab et al. (2018) investigated the effect of digital-based learning on the knowledge and attitude of 150 high school teachers about the prevention of health-risk behaviors in students. The result showed that both multimedia and booklets could enhance the teachers’ knowledge and improve their attitudes toward the prevention of health-risk behaviors [28]. Chu et al. (2019) examined the ability of new nurses to conduct pain assessment and treatment through multimedia and simulated scenario instruction. In this study, 86 new nurses were allocated into two groups: the experimental group and the control group. The control group underwent pain assessment training by traditional slides in classrooms. As for the experimental group, participants received the training with the same contents as that in the control group but in a multimedia-assisted model. Post-test questionnaires were completed after training, and the results showed that the experimental group had significantly higher satisfaction scores and demonstrated greater knowledge and pain assessment ability than the control group [29]. The multiple media courses are able to demonstrate practical and skill lectures completely, and could be an important factor that makes students deepen learning impression and internalize the contexts into their own knowledge. In digital-based learning, the learning profile, completion, and result may be recorded through sound learning platforms such as viewing times and duration of watching; the instructors may evaluate the learning process and outcomes [30]. However, it is worthy to consider that the changes of attitude and behavior are difficult to measure. In this study, even though the behaviors between two groups did not reach statistical significance in the regression analysis, there were significant differences in knowledge and attitude between groups. The literature indicated that behavior changes usually happen after knowledge and attitude have changed. This study only evaluated the learning outcomes pre- and post- lecture, therefore, the short-term effects of changes in knowledge and attitudes were observed. In terms of long-term influences of behavior changes, more time is needed to complete the complete evaluation. 

This study showed similar results with the studies mentioned above, in those studies conducted in Taiwan and overseas, multimedia learning could improve knowledge, attitude, and behavioral intention [24,28,29]. However, Lahti, Hätönen, and Välimäki (2014) had a different finding in their systematic review study; there were 2491 nurses and nursing students were enrolled. They concluded that there was no statistically significant difference between groups in e-learning and traditional learning relating to nurses’ or nursing students’ knowledge and skills [31]. This may be due to the fact that skills in the digital-based learning had not been integrated into clinical practice in the past, which may lead to no difference between digital learning and traditional classroom learning. The learning material in this study is about infection control skills that are urgently needed and closely related to the participants; thus, the learning outcome of digital-based learning was improved largely.

In terms of the associated factors, the extent of knowledge improvement was significantly associated with the experience of providing care for cases with COVID-19. The score improvement in the attitude was associated with work experience, where participants with at least ten years of working experience had a greater extent of score improvement in the attitude than those who have worked for less than three years [32].

According to the results of the literature reviewed above and the research results in this study, digital-based educational intervention exhibits greater learning outcomes in knowledge, attitude, and behavioral intention in nurse aides than traditional face-to-face learning. It is suggested that digital-based learning can be used as an efficient method of educational training for nurse aides. Especially during the pandemic of COVID-19, digital-based learning may decrease the risk of cluster infection and community infection [33]. In the rapid-change circumstances under the pandemic, multimedia-based learning is strongly suggested to be included in nurse aides’ continuous education in the future; in order to increase the acquisition of pathways, reducing the impact factors from outside environment to complete the in-service education is needed. Digital learning could save learners’ time and costs in updating the latest knowledge and skills [34].

## 5. Limitations

Under the pandemic, this study only enrolled nurse aides from one medical center in central Taiwan; thus, the result may not be generalized to all the nurse aides in hospitals. Every hospital has its own training system and nursing guidelines which could be very different; this study only explored nurse aides’ in-service education program in one medical center. The generalization of the study results could be improved if the same education program is applied in various sites with larger sample size, or in other professional groups who work in hospital. 

Although only a small number of participants had limited experiences in caring forCOVID-19 patients, this educational program is still worthy of being applied in training new nurse aides, because nurse aides can immediately learn the knowledge, attitude and skills, and apply them to care for patients immediately after this education program. In this multimedia-based learning design, there were no interactive activities included in the course, which might limit the learning effectiveness. In future multimedia-based learning course design, interactive learning model should be introduced to increase the learning effects. 

## 6. Conclusions

This study demonstrates that multimedia-based educational intervention can deliver knowledge and skills related to COVID-19. In addition, the video display may be helpful in reinforcing the impression of learning in nurse aides. Therefore, during a global pandemic with such great impact, a multimedia-based learning approach should be considered as an effective strategy to avoid cluster infection, enhance knowledge acquisition, and strengthen learners’ attitude and behavioral intention simultaneously. However, the effectiveness and applicableness of multimedia-based learning should be further evaluated in different areas, populations, and professional groups.

## Figures and Tables

**Figure 1 healthcare-10-01206-f001:**
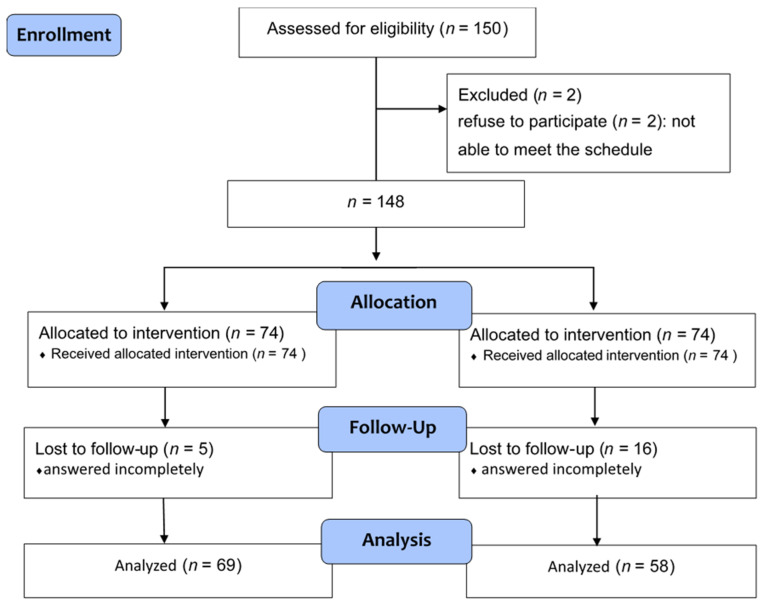
Participant recruitment and follow-up flowchart.

**Table 1 healthcare-10-01206-t001:** Distributions of demographic characteristics of the study participants.

Variables	Total (*n* = 127)		Multimedia-Based Learning (*n* = 69)		Face-to-Face Learning (*n* = 58)		*p*-Value
	n	%	n	%	n	%	
Gender							0.716
Male	32	25.2	16	23.2	16	27.6	
Female	95	74.8	53	76.8	42	72.4	
Age (years)							0.013
50 or under	26	20.5	8	11.6	18	31.0	
Over 50	101	79.5	61	88.4	40	69.0	
Marital Status							0.107
Married	103	81.1	60	87.0	43	74.1	
Single	24	18.9	9	13.0	15	25.9	
Education Level							0.209
Junior high school	47	37.0	26	37.7	21	36.2	
Senior high school	65	51.2	38	55.1	27	46.6	
University/Junior college	15	11.8	5	7.2	10	17.2	
Working Experience							<0.001
Under 3 years	18	14.2	3	4.4	15	25.9	
3 to 10 years	58	45.6	41	59.4	17	29.3	
Over 10 years	51	40.2	25	36.2	26	44.8	
Experience of Caring for a Case with Infectious Diseases							0.016
Yes	79	62.2	50	72.5	29	50.0	
No	48	37.8	19	27.5	29	50.0	
Experience of Caring for a Case with COVID-19							0.042
Yes	5	3.9	0	0.0	5	8.6	
No	122	96.1	69	100.0	53	91.4	

**Table 2 healthcare-10-01206-t002:** Distributions and comparisons of the scores in knowledge, attitude, and behavioral intention.

Item	Multimedia-Based Learning (*n* = 69)		Face-to-Face Learning (*n* = 58)		
	Mean ± SD		Mean ± SD		*p*-Value ^†^
	Pre-test	Post-test	Pre-test	Post-test	
Score of Knowledge	4.94 ± 1.57	9.48 ± 0.63 ***	7.00 ± 2.41	8.72 ± 1.54 ***	<0.001
Score of Attitude	38.54 ± 4.56	62.84 ± 2.13 ***	31.78 ± 9.71	44.29 ± 12.95 ***	<0.001
Score of Behavioral Intention	4.17 ± 1.26	5.91 ± 0.28 ***	5.16 ± 1.24	5.90 ± 0.31 ***	<0.001

*** *p* < 0.001 based on the paired t-test for comparing the mean scores between pre-test and post-test. ^†^
*p*-value of the interaction term (group*time) in the generalized estimating equations.

**Table 3 healthcare-10-01206-t003:** Correlation analysis for the score change in knowledge, attitude, and behavioral intention.

Item	Score Change in Knowledge	Score Change in Attitude	Score Change in Behavioral Intention
Score Change in Knowledge	1		
Score Change in Attitude	0.288 **	1	
Score Change in Behavioral Intention	0.534 **	0.235 **	1

** *p* < 0.01.

**Table 4 healthcare-10-01206-t004:** Multiple linear regression analysis: exploring associated factors for the score changes in knowledge, attitude, and behavioral intention.

	Score Change in Knowledge		Score Change in Attitude		Score Change in Behavioral Intention	
	Regression Coefficient	*t*	Regression Coefficient	*t*	Regression Coefficient	*t*
Gender						
Male	ref.		ref.		ref.	
Female	−0.422	−0.923	−0.473	−0.225	0.337	1.377
Age						
50 or under	ref.		ref.		ref.	
Over 50	−0.661	−1.856	−2.047	−1.236	0.144	0.740
Marital Status						
Married	ref.		ref.		ref.	
Single	−0.001	−0.003	−3.321	−1.583	−0.165	−0.670
Education Level						
Junior high school	ref.		ref.		ref.	
Senior high school	−0.322	−0.735	−1.283	−0.638	−0.218	−0.930
University/Junior college	−1.644	−2.375 *	0.966	0.297	−0.471	−1.246
Working Experience						
Under 3 years	ref.		ref.		ref.	
3 to 10 years	−0.384	−0.488	8.300	2.298	−0.135	−0.315
Over 10 years	0.294	0.388	8.213	2.362 *	0.437	0.293
Experience of Caring for a Case with Infectious Diseases						
Yes	ref.		ref.		ref.	
No	−0.184	−0.411	−1.559	−0.759	0.445	1.857
Experience of Caring for a Case with COVID-19						
Yes	ref.		ref.		ref.	
No	−3.918	−3.588 ***	4.219	0.799	0.339	0.551
Group						
Multimedia-based Learning	ref.		ref.		ref.	
Face-to-face Learning	−3.086	−7.085 ***	−11.946	−4.997 ***	−0.256	−0.836
Score Change of Knowledge			0.314	0.737	0.239	4.813 ***
Score Change of Attitude					0.007	0.603

* *p* < 0.05, *** *p* < 0.001.

## Data Availability

Not applicable.

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
