# Peer review of "Effectiveness of Multimedia-Based Learning on the Improvement of Knowledge, Attitude, and Behavioral Intention toward COVID-19 Prevention among Nurse Aides in Taiwan: A Parallel-Interventional Study"

_healthcare, 2022, doi:10.3390/healthcare10071206_

Round 1

Reviewer 1 Report

This study aimed to explore and compare the learning outcomes between the digital-based learning and traditional face-to-face learning regarding the knowledge, attitude and behavioral intention of nurse aides. This study demonstrates that digital-based educational intervention can delivery knowledge and skills related to COVID-19. The study is interesting, but I have some comments.

TitIe. I suggest included the study location and Design.

The summary should be improved is very poor. I will not include results or conclusion.

Introduction. It must be improved. Contextualize to digital-based learning in improving knowledge, attitude and behavioral

Methods.

Include study design.

Describe the location and area of the study in more detail.

Include the flow chart of the CONSORT guide.

How did you calculate the sample?

Results.

Table 1. Remove the chi-square, only consider the p-value.

Table 4. I suggest changing the analysis. Include a crude OR and an OR adjusted for confounding factors.

Discussion.

Describe the public health implications of the study.

Author Response

This study aimed to explore and compare the learning outcomes between the digital-based learning and traditional face-to-face learning regarding the knowledge, attitude and behavioral intention of nurse aides. This study demonstrates that digital-based educational intervention can delivery knowledge and skills related to COVID-19. The study is interesting, but I have some comments.

TitIe. I suggest included the study location and Design.

Ans.

Thank you for your advice. We have added the study location and design in the title.

The summary should be improved is very poor. I will not include results or conclusion.

Introduction. It must be improved. Contextualize to digital-based learning in improving knowledge, attitude and behavioral

Methods.

Include study design.

Describe the location and area of the study in more detail.

Include the flow chart of the CONSORT guide.

How did you calculate the sample?

Ans.

Thank you for your suggestion. We have added the flow chart, sample size, in the method section.

Results.

Table 1. Remove the chi-square, only consider the p-value.

Table 4. I suggest changing the analysis. Include a crude OR and an OR adjusted for confounding factors.

Ans.

Thank you for your suggestion. We have removed the chi-square in table 1. In terms of table 4, we conducted multiple regression analysis to understand whether the demographic variables influence the pre and post-test in knowledge, attitude and behavior; therefore, there is no OR to be presented.

Discussion

Describe the public health implications of the study.

Ans.

Thank you for your advice. We have added some public health implications.

Reviewer 2 Report

The work "Digital-based Learning in Improving Knowledge, Attitude and Behavioral Intention toward COVID-19 in Nurse Aides" is very timely. The objective of the work is clear and the results are seemingly interesting and important for policy toward future training during pandemics.

However, the good work suffers a lot of setbacks.

The introduction is shallow, could be improved to make your work of more interest to the scientific audience.

The sample size of the work is very small given the importance this study serves. If you could add on it would be better.

The references are very insufficient for such as study. I advise you to consult up to more than 30 0r 40 studies to help you sharpen your hypothesis as well as discussion your results. With such shallow referencing, it undermines the scientific muscle of your work.

Author Response

The work "Digital-based Learning in Improving Knowledge, Attitude and Behavioral Intention toward COVID-19 in Nurse Aides" is very timely. The objective of the work is clear and the results are seemingly interesting and important for policy toward future training during pandemics.

However, the good work suffers a lot of setbacks.

The introduction is shallow, could be improved to make your work of more interest to the scientific audience.

The sample size of the work is very small given the importance this study serves. If you could add on it would be better.

The references are very insufficient for such as study. I advise you to consult up to more than 30 0r 40 studies to help you sharpen your hypothesis as well as discussion your results. With such shallow referencing, it undermines the scientific muscle of your work.

Ans.

Thank you for your suggestion to make this paper more solid. We have updated some references in introduction and discussion parts. Also, we have added the sample size inside method.

Reviewer 3 Report

The article presents the results of a study that aims to compare the expansion of knowledge and improvement of skills related to COVID-19 between digital-based educational sessions and traditional face-to-face learning models for nurse aides.

Participants in this study are nurse aides who worked in a medical center in central Taiwan. They were divided into two groups: one of them received digital-based teaching, and the other received traditional face-to-face teaching. Participants of these two groups were asked to complete a pre-test questionnaire before the session and a post-test questionnaire.

The study results show the great impact of the digital-based learning approach and it argues that it should be considered as an effective strategy to enhance knowledge acquisition and avoid cluster infection.

Despite the exaltation of the digital-based learning approach related to COVID-19, the research approach used by the study provides little indication both on the context conditions and on the specific contents of the training that could have determined the success of the digital-based learning approach. The result is that the current study, despite its exaltation of the results of the digital-based learning approach, provides little contribution to understanding the elements that can favor its success.

The limited dialogue the study has with the most recent literature on the subject does not help either.

Author Response

The article presents the results of a study that aims to compare the expansion of knowledge and improvement of skills related to COVID-19 between digital-based educational sessions and traditional face-to-face learning models for nurse aides.

Participants in this study are nurse aides who worked in a medical center in central Taiwan. They were divided into two groups: one of them received digital-based teaching, and the other received traditional face-to-face teaching. Participants of these two groups were asked to complete a pre-test questionnaire before the session and a post-test questionnaire.

The study results show the great impact of the digital-based learning approach and it argues that it should be considered as an effective strategy to enhance knowledge acquisition and avoid cluster infection.

Despite the exaltation of the digital-based learning approach related to COVID-19, the research approach used by the study provides little indication both on the context conditions and on the specific contents of the training that could have determined the success of the digital-based learning approach. The result is that the current study, despite its exaltation of the results of the digital-based learning approach, provides little contribution to understanding the elements that can favor its success.

The limited dialogue the study has with the most recent literature on the subject does not help either.

Ans.

Thank you for your suggestion to make this paper more solid. We added detailed background of digital-based learning in introduction, and more details in discussion. Hope these changes could make our points more clearly.

Reviewer 4 Report

Although I see some merit in the actual contents of the paper, paper needs great revision in contents. In its current form, I remain skeptical of the advantages that the proposal provides.

 1.              Introduction. This section should be improved. Introduction section is long, but vague and weak. Introduction should show paper motivation, paper purpose and which is the paper knowledge contribution. After reading it, the research objectives and their importance are hidden. Authors should improve introduction section showing better (1) the problem that they are trying to solve, (2) the paper objectives, and (3) justifying why their proposal is necessary and its benefits. Paper motivations could be considerably strengthened by providing evidence, in practice and in theory, as to why is necessary to develop this proposal. The paper, as it currently stands, isn’t strongly motivated in terms of how it meets an existing gap in the literature.  

 2.              A literature review section is necessary. Describe here all the concepts that are going to use in the paper.

3.              Discussion. The contribution of the author’s approach to the literature is not highlighted. The literature review needs to be integrated with the claims that the author make in order to show the importance of its contribution. 

In the discussion section include the summary of the findings and highlight the contribution of your study to the literature. Authors should compare their findings with other works in the literature, standing out what their contribution to the State of the art is, and if the findings fit with what was expected. 

Overall, try to provide sufficient validation regarding the novelty of this research along with beneficial.

In addition, implications for managers and practitioners, and academics, should be highlighted in the discussion section.

4.              Conclusion. Begin with a brief summary of paper motivations, objectives and findings. Finally, include here limitations and future research. I would like that authors improve the limitations  and propose more future research

Author Response

 Although I see some merit in the actual contents of the paper, paper needs great revision in contents. In its current form, I remain skeptical of the advantages that the proposal provides.

  1. Introduction. This section should be improved. Introduction section is long, but vague and weak. Introduction should show paper motivation, paper purpose and which is the paper knowledge contribution. After reading it, the research objectives and their importance are hidden. Authors should improve introduction section showing better (1) the problem that they are trying to solve, (2) the paper objectives, and (3) justifying why their proposal is necessary and its benefits. Paper motivations could be considerably strengthened by providing evidence, in practice and in theory, as to why is necessary to develop this proposal. The paper, as it currently stands, isn’t strongly motivated in terms of how it meets an existing gap in the literature.  
  2. A literature review section is necessary. Describe here all the concepts that are going to use in the paper.

Ans.

Thank you for your suggestion to make this paper more solid. We added detailed background and updated references of digital-based learning in introduction. We hope these changes could both introduction and literature review more fit to this research and draw our points more clearly.

  1. Discussion. The contribution of the author’s approach to the literature is not highlighted. The literature review needs to be integrated with the claims that the author make in order to show the importance of its contribution. 

In the discussion section include the summary of the findings and highlight the contribution of your study to the literature. Authors should compare their findings with other works in the literature, standing out what their contribution to the State of the art is, and if the findings fit with what was expected. 

Overall, try to provide sufficient validation regarding the novelty of this research along with beneficial.

In addition, implications for managers and practitioners, and academics, should be highlighted in the discussion section.

Ans. Thanks for the instruction of how to improve the discussion. We have made some changes to integrate our research findings and made comparisons with other research.

  1. Conclusion. Begin with a brief summary of paper motivations, objectives and findings. Finally, include here limitations and future research. I would like that authors improve the limitations and propose more future research

Ans. Thank you for the detailed suggestion. We tried our best to address these points on time.

Round 2

Reviewer 1 Report

No comments

Reviewer 2 Report

The authors improved a bit but the paper is still lacking scientifically.

As i said, you need to consult more work and improve your discussion of the study findings.

Your work is good but can not be recommended for publication in this state, look at my previous comments. 

Look at some previous papers both on the topic and also the journal to not only work on the scientific part but also the journal format.

Author Response

Reviewer 2

Comments and Suggestions for Authors

The authors improved a bit but the paper is still lacking scientifically.

As i said, you need to consult more work and improve your discussion of the study findings.

Your work is good but can not be recommended for publication in this state, look at my previous comments. 

Look at some previous papers both on the topic and also the journal to not only work on the scientific part but also the journal format.

Ans.

Thank you for these important advises. We tried added more previous research paper results to enhance and consolidate the discussion to be more scientific.

Reviewer 3 Report

Digital learning, as a learning method, is currently the subject of a rich multidisciplinary scientific research. Inside, it is possible to find a careful analysis of its potential and limits, with attention to the technological and non-technological conditions, which can favor or limit its implementation and its success. This article remains outside the debate on the subject. It appears to be concerned with the affirmation of the success of digital learning, not placing the study in the international debate. This does not contribute to underlining the relevance of the work carried out or to discussing its results.

The objective of the article is to present the results of a study that compares the improvement of knowledge and of skills related to COVID-19 between two groups of nurse aides at a medical center in central Taiwan. One group attended digital-based educational sessions and the other traditional face-to-face learning models.

The main concern of the study appears to have been evaluation of the efficacy of digital learning and its success; however, it is still difficult to grasp the scientific value of the article, which does not substantially engage with debates on the subject.

In this regard I would like to recall that digital learning, as a learning method, is currently the focus of a rich, multidisciplinary scientific research. From this work, it is possible to find careful analysis of its potential and limits, with reference to technological and non-technological conditions, which can favor or limit its implementation, and its success.

It would have been appropriate for the authors to have entered into dialogue with this literature. This would have contributed to a more innovative realization of study objectives.

In the absence of this, it would be advisable to try to put the results of the study in a framework informed by national and international literature on the subject. This means, among other things, worrying less about exalting the value of digital learning, and more about trying to grasp its potential and its limits, taking into account multiple variables.

Author Response

(The authors gave the same response as above.)

Reviewer 4 Report

Authors have made a good work. In my opinion, paper is now ready for publication
